# The Effect of a Gluten-Free Diet on Vitamin D Metabolism in Celiac Disease: The State of the Art

**DOI:** 10.3390/metabo13010074

**Published:** 2023-01-02

**Authors:** Michele Di Stefano, Emanuela Miceli, Caterina Mengoli, Gino Roberto Corazza, Antonio Di Sabatino

**Affiliations:** 11st Department of Medicine, IRCCS “S.Matteo” Hospital Foundation, 27100 Pavia, Italy; 2Department of Internal Medicine, University of Pavia, 27100 Pavia, Italy

**Keywords:** celiac disease, gluten-free diet, vitamin D, hyperparathyroidism

## Abstract

Celiac disease is a chronic autoimmune disorder involving the small intestine, characterized by villous atrophy, crypt hyperplasia and an increase in intraepithelial lymphocytes. Due to both calcium malabsorption and immune activation, a high prevalence of bone mass derangement is evident in this condition, regardless of the presence of overt malabsorption. Alterations of mineral metabolism are also frequently described, and in this review, the modifications of serum levels of vitamin D are analyzed, according to the available literature on this topic. In untreated patients, secondary hyperparathyroidism is responsible for the hyperconversion of 25-vitamin D into 1,25-vitamin D making mandatory the determination of serum levels of both vitamin metabolites to avoid a wrong diagnosis of vitamin D deficit. A gluten-free diet allows for a normalization of bone and mineral metabolism, reverting these abnormalities and raising some doubts on the need for vitamin supplementation in all the patients. Data available do not support this wide indication, and a complete evaluation of bone and mineral metabolism should be performed to select patients who need this therapeutic approach.

## 1. Introduction

Celiac disease (CD) is a chronic autoimmune disorder of the small intestine caused by the intake of gluten in genetically predisposed subjects. The gluten-evoked immune response alters enteric mucosa architecture, determining villous atrophy, crypt hyperplasia and an increased number of intraepithelial lymphocytes [1]. Mucosal lesions compromise the absorbing capacity due to a reduction in intestinal surface area and cause, among others, calcium malabsorption and vitamin D metabolism alterations [2,3]. Together with the persistent inflammation [4], intestinal malabsorption is considered a pivotal mechanism for bone mass and mineral metabolism impairment. The impairment of calcium absorption and the consequent endocrine alterations of the secretion of parathyroid hormone and vitamin D metabolism deeply modify bone homeostasis, causing an increased bone resorption stimulating bone turnover. In this review, we have analyzed the evidence describing the impairment of bone and mineral metabolism, and we have suggested both a correct timing for their measurement and a correct strategy to select patients who need a vitamin D supplementation along their life. This is still an unmet need, as guidelines for the management of osteoporosis in CD patients lack specific suggestions on bone metabolism alteration approaches [5,6].

## 2. Bone and Mineral Metabolism in CD Patients at Diagnosis and after GFD

It is estimated that three-quarters of patients with overt malabsorption at diagnosis and half of patients with minor symptoms suffer from significant alterations of bone mineralization [7], but alterations of calcium metabolism are present in all the patients. In comparison with CD patients at diagnosis, the prevalence of bone loss in patients following a strict GFD is significantly lower [7], suggesting a therapeutic role of GFD on this consequence of CD. It is known that fracture risk is significantly increased in postmenopausal women [8], and when bone loss is present, it is very difficult to induce its reversal with drug treatment. Therefore, the importance of bone loss prevention, through an active search for bone derangement in all patients suffering from conditions responsible for a secondary osteoporosis, is evident. In CD patients diagnosed during childhood, a strict GFD from childhood normalizes BMD [9], and a significant bone mass gain could be evident, even after a six-month period of GFD [10]. Unfortunately, this is not the same in CD patients diagnosed in adulthood [7], making the adoption of strategies aimed at the prevention of osteoporosis very important.

Calcium represents one of the main regulators of muscular contraction, and its serum levels are finely monitored and balanced to support cardiac muscle activity. Unless severe malabsorption symptoms characterize the disease at diagnosis [11], reduced serum calcium levels are unfrequently detected in untreated CD patients. The increased attention to CD diagnosis and the adoption of screening tests for first-degree relatives of CD patients elicits the recognition of patients in an early phase of the disease, generally characterized by minor symptoms caused by subclinical malabsorption and a low degree of severity of bone derangement. Parathyroid hormone (PTH) and vitamin D represent the main actors of the regulation of calcium metabolism, and their serum level alterations in untreated CD patients suggest that calcium malabsorption is a pivotal mechanism in the pathophysiology of bone loss. All the papers dealing with bone mass and mineral metabolism in untreated CD patients agree with this issue.

Vitamin D is a secosteroid that is detectable in blood in two main forms: 25-hydroxyvitamin D, that is, 25(OH)-vitamin D, an indicator of the entity of vitamin D storage; and 1,25-dihydroxyvitamin D, that is, 1,25(OH)_2_-vitamin D, the active metabolite, regulating calcium balance and modulating the immune system.

In untreated CD patients, a complex network of events is arranged to rapidly compensate serum calcium reduction and prevent fecal losses: in Figure 1, the cascade of endocrine reaction to hypocalcemia in untreated CD patients is shown [7].

The release of PTH and its effect on the kidney, stimulating 1-α-hydroxylase activation, and in turn, enhancing the conversion of 25-vitamin D in 1,25-vitamin D with the consequent effect of activated vitamin D on intestinal calcium absorption through an increased vitamin D-dependent active transport, is evident. Even if enterocytes express a normal number of vitamin D receptors [3], negligible cytoplasmatic levels of vitamin D-dependent calcium-binding protein at enterocyte level [12] frustrate such a compensatory effort, and in fact, the activation of the complex network expresses only its negative effects on bone mineralization, played by the direct proresorptive effect of increased levels of both PTH and 1,25-vitamin D [13].

Accordingly, in untreated CD, serum levels of 25 vitamin D are very frequently low; on the contrary, serum levels of 1,25-vitamin D are frequently high. Many papers have clearly shown the characteristic behavior of these two metabolites. High levels of 1,25-vitamin D at diagnosis are frequently associated with high levels of PTH [4,14], and a strict GFD generally normalizes these alterations [4,14,15,16,17]. GFD-related improvement of PTH levels seem strictly related to the reversal of mucosal lesions, as hyperparathyroidism persists both in patients with refractory CD [10] and patients with a low compliance to GFD [18], and is associated with alterations of vitamin D metabolite levels [16]. However, one year of GFD duration is not always sufficient to normalize PTH levels [18], while a longer period was clearly shown to be effective [4,14,15,16,19]. It is worth noting that in patients with CD at diagnosis, the previous use of calcium and vitamin D supplements may be effective in the prevention of hyperparathyroidism [20]: on clinical grounds, this is an important issue, as these pivotal alterations of bone and mineral metabolism may be masked by the supplementation. Physicians could be misled.

In untreated patients with CD, circulating levels of the active metabolite of vitamin D show a pattern of modification similar to PTH. In the presence of high levels of PTH, due to its effect on renal 1-α-hydroxylase, high levels of 1,25-vitamin D are expected, and many papers confirm this issue. In particular, it was previously shown that serum levels of PTH and 1,25-vitamin D are strictly correlated even in patients on long term GFD [21]. In patients on GFD for a median 2-year period, we showed significantly lower levels of serum levels of 1,25-vitamin D in comparison with patients at diagnosis [14,22]. As expected, these results were confirmed in patients on GFD for longer periods of time [4,16,19] and the duration of GFD seems crucial for the complete normalization. More interestingly, it was shown that the reduction of serum levels of 1,25-vitamin D is very rapid, beginning after just three months of GFD [23]. 

The reversal of altered calcium balance also reflects its effects on 25-vitamin D serum levels, which increase after the introduction of GFD. In a cohort of 44 CD patients on long-term GFD, we recently showed a normalization of 25-vitamin D levels, independently from the presence of bone loss [24]. This is a largely confirmed result: all available papers agree on this issue. Low levels of 25-vitamin D in untreated CD could be caused by a reduction in intestinal absorption, a reduction in dietary intake and a reduction in exposure to sunlight [25]. However, a reduction in 25-vitamin D half-life in patients with malabsorption was also shown [26].

Ultimately, the presence of vitamin D deficiency in untreated CD patients should be considered very rare, as 25-vitamin D levels are low, but are always accompanied by high levels of 1,25-vitamin D. On the contrary, the presence of vitamin D deficiency in treated CD patients represents a more important condition on clinical grounds, deserving a different diagnostic and therapeutic approach. Unfortunately, very few papers report the prevalence of vitamin D deficiency in treated adult CD patients. On the basis of previous results, the prevalence of 25-vitamin D deficiency is expected to decrease following a strict GFD, and this was evident, in particular, in a 5-year longitudinal follow-up study on a cohort of untreated patients: the prevalence of 68% at baseline was 54% after one year of GFD and fell down to 8% after 5 years of diet [26].

It is, however, conceivable that both age at diagnosis of CD and age at the time of vitamin D measurement may condition the modification of vitamin D levels during the CD patient’s lifespan, but we have no information on this issue. Recent papers reported a prevalence of 25-vitamin D deficiency in treated CD patients of 25% [27] and 28% [28], but we have no information on the age at measurement and on the duration of GFD of these subgroups of patients. Only one study, describing a prevalence of 41%, reported the absence of a significant difference between patients with and without 25-vitamin D deficiency [29]. A critical point is, therefore, the discrepancy between the frequent normalization of serum indices of bone and mineral metabolism and the difficult normalization of BMD levels in treated CD, even after long periods of GFD.

Several studies suggested a role of vitamin D receptor genotypes in the pathophysiology of both bone loss and vitamin D serum alterations in patients with CD [30,31,32]. Among the high number of single-nucleotide polymorphisms (SNPs), Fok I (SNP rs 2228570), Bsm I (SNP rs1544410), Apa I (SNP rs7975232) and Taq I (SNP rs731236) are the most studied. A recent meta-analysis excluded the causal relationship between VDR polymorphism and the alteration of bone and mineral metabolism [33].

## 3. Awareness of the Persistence of Nutritional Deficiencies in CD Patients following GFD

GFD represents the treatment of CD, but both in children [34] and in adults [35,36], it was shown that it is not sufficient to prevent nutritional deficiency. Recently, the adoption of GFD was considered a very healthy measure for patients affected by a series of conditions different from CD [37] as well as for healthy subjects, according to an unproven hypothesis attributing the ability to guarantee wellness to gluten-free products [38]. 

Di Nardo et al., in their systematic review dealing with nutritional deficiencies in childhood CD, showed an increased risk of excessive fat and an insufficient intake of vitamin D, besides fibers, iron and calcium [34]. The daily intake of vitamins B1, B2 and B6 as well as folate proved to be significantly lower in CD on GFD than in the general German population [39]. Vitamin B deficiencies should be considered secondary to the ingestion of gluten-free cereals containing lower amounts of folate than gluten-containing cereals [40]. It should be also considered that the folic acid content of quinoa is 78.1 mg/100 g, while amaranth contains 102 mg/100 g, which is substantially higher than wheat, containing an level of folate of 40 mg/100 g [41]. It was recently shown that in comparison with gluten-containing products, gluten-free equivalents are characterized by lower levels of vitamin D, vitamin E, vitamin B12, folate, iron, magnesium, potassium and sodium [42]. Similar considerations could be made for gluten-free breads: only 1 product out of 20 tested was proven to be fortified with iron, calcium, thiamin and niacin, while 3 out of 10 were fortified only with calcium and iron. The intake of minerals in CD patients on GFD was shown to be insufficient, in particular for calcium, iron, manganese and zinc [40], due to the low mineral concentration of these minerals in gluten-free raw materials. It was recently shown that mineral deficiency in CD patients on a long-term GFD is detectable in 10% of patients, in particular magnesium and calcium for both males and females, as well as iron for females and zinc for male patients [43]. Apart from iron deficiency, the presence of other mineral deficiencies is frequently underestimated in CD patients, and a series of metabolic consequences could be lost in the clinical follow up. For instance, this is the case of zinc, as its deficiency may impair protein synthesis and hamper growth [44]. 

From these studies clearly emerges the information that even if patient adherence to GFD is judged to be perfect, the lack of fortification increases the risk of micronutrient deficiency. Accordingly, follow-up visits of CD patients should be also aimed at the prevention, or at least at the early detection, of these complications.

Recently, among both healthcare professionals and medical students, an assessment of the knowledge of the risk of nutritional deficiencies in CD patients following a strict GFD was performed [45]. Results showed that only 37% of interviewed healthcare professionals provided at least 60% of correct answers, suggesting a very low level of awareness of the long-term risk of nutritional deficiencies in CD patients, strictly adhering to GFD. These results underline the need for ad hoc educational programs finalized to a deep learning of the nutritional requirements of CD patients in their real life.

Moreover, the need for an expert dietitian in monitoring GFD in CD was recently evaluated [46]. This is a very important issue, as the replacement of wheat, rye and barley with gluten-free equivalents or grains such as maize, millet, rice, teff and tapioca may expose CD patients to an increased intake of fibers and an increased intake of fat [47]; this different diet increases the risk of metabolic disorders such as obesity, diabetes, hypertension and cardiovascular complications. An increased fat intake may expose patients to a reduced absorption of magnesium [48]. The increased intake of fibers in CD patients may also contribute to the persistence of gas-related symptoms, such as bloating, flatulence and abdominal distention [49], evoking the coexistence of other disorders, in particular functional bowel disorders. Accordingly, the role of dietitians in the follow-up of CD patients should be emphasized and strongly suggested in order to prevent the occurrence of gas-related symptoms and perform a sort of direction of the diet. 

## 4. When Is It Correct to Supplement Vitamin D in CD Patients?

There are many concerns about nutritional supplementation in CD patients, and guidelines are frequently characterized by substantial differences. A recent Consensus Conference underlined the need for further studies aimed at the clarification of the role of vitamin D in the pathophysiology of CD and the need for supplementation [50]. The case of vitamin D is, however, not isolated, as the same considerations could be expressed for calcium supplementation. A correct intake of calcium for the maintenance of optimal bone health and a balance of mineral metabolism depends on the age of the subjects. Adolescents and old subjects need a high amount of calcium due to the rapid growth and the reduced intestinal calcium absorption capacity, respectively. However, the Food and Agriculture Organization (FAO) recommends a calcium intake of 800–1000 mg/day in men and women over 50 years of age [51]. The Institute of Medicine of the United States National Academy of Sciences suggests 1000 mg for 19–50-year-old women and 19–70-year-old men and 1200 mg/day for postmenopausal women and men over 70 years old [52]. The increase in calcium supplementation to over 1200 mg/day may predispose one to an increased risk of urinary stones, but the risk of this complication should be monitored in all the subjects following a pharmacological calcium supplementation.

Serum levels of 25-vitamin D below the value of 30 nmol/L are considered correlated to an increased risk of bone derangement, and values higher than 50 nmol/L or 20 ng/mL are considered appropriate for bone health [53,54,55,56]. A lack of standardization is, however, evident for vitamin D measurement, and comparisons between methods have reported discrepancies of immunoassays in comparison with LC-MS/MS methods [57]. This is still an unmet need hampering clinical research: all the efforts to overcome this impasse are welcome. 

An appropriate vitamin D supplementation should consider age and gender, but also skin type, season and geographic area. The USA Institute of Medicine and the European Food Safety Authority (EFSA) agreed to consider 15 microg of vitamin D as a recommended dietary allowance (RDA) [51,57]. In particular, expressed in IU/day, an RDA of 600 IU/day is suggested for adults in the age range of 50–70 years and 800 IU/day is suggested for adults aged >70 years. The goal of vitamin D supplementation should be the achievement of serum 25 vitamin D levels of 50 nmol/L, an effective value for bone health maintenance in more than 97% of North America’s people [58,59]. 

### 4.1. Vitamin D Supplementation in Untreated CD Patients

As far as CD patients are concerned, the increased conversion of 25-vitamin D into 1,25-vitamin D is a crucial point and should be particularly stressed. In untreated CD patients, low levels of 25-vitamin D should not be necessarily interpreted as the expression of vitamin deficiency: the hyperconversion to 1,25-vitamin D, secondary to hyperparathyroidism, is the cause of low 25-vitamin D. Consequently, 25-vitamin D supplementation in CD patients at diagnosis could not only result in a useless therapeutic measure, but even in a dangerous treatment, as excessively elevated serum levels of 1,25-vitamin D are characterized by a proresorptive effect, as they enhance bone loss [60]. 

The indication of vitamin D supplementation in untreated adult CD patients should arise from the evaluation of circulating levels of both vitamin D metabolites, together with the measurement of PTH levels. Low levels of 25-vitamin D associated with high levels of 1,25-vitamin D and PTH should be not approached with supplementation of drugs aimed at the increase in 25-vitamin D, as this measure may enhance the already-stimulated hyperconversion to 1,25-vitamin D and may increase the risk of 1,25-vitamin D-mediated bone resorption. Moreover, it was clearly shown that 25-vitamin D supplementation as an add-on therapy to GFD does not improve bone mass gain in comparison with GFD alone [23], making this therapeutic measure even more useless.

On the contrary, two different approaches could be suggested. Due to the rapid reversal of GFD-mediated modification of vitamin D metabolite levels [23], a conservative approach is justified, associated with a re-evaluation of serum levels of 25-vitamin D, 1,25-vitamin D and PTH after 3 months of GFD [22], to check for the positive effect of the diet. 

An alternative approach deduces its rationale on the presence of calcium balance impairment. In untreated CD, the modification of vitamin D and PTH serum levels is the expression of the impaired calcium balance independently from the actual detection of hypocalcemia. Accordingly, oral supplementation of calcium could represent a useful, preliminary therapeutic approach. Villous atrophy may hamper the efficacy of this measure, as untreated CD patients show a reduced calcium absorption, which improves on GFD [14]. However, the description of a rapid improvement of serum levels of both 25-vitamin D and 1,25-vitamin D after three months of GFD [23] suggests that the effect of calcium supplementation on the improvement of secondary hyperparathyroidism is low at the beginning of GFD. Instead, the effect could be evident within the third month of the diet. Moreover, the presence of normal levels of vitamin D and PTH during calcium and vitamin D supplementation begun before diagnosis in untreated CD patients suggests a positive effect of this therapeutic measure, besides the presence of villous atrophy [20].

### 4.2. Vitamin D Supplementation in Treated CD Patients

In patients with treated CD, the need for vitamin D supplementation seems less urgent, especially in those patients showing a progressive normalization of biomarkers of calcium balance after the beginning of GFD. If patients follow a strict GFD, it is very improbable that alterations of calcium balance may originate from mechanisms related to gluten toxicity. However, recently, nutritional and biochemical parameters of a group of CD patients on GFD recruited through a patient’s association were compared to a group of non-celiac adult volunteers [61]. In both the studied groups, vitamin D intake did not reach the recommended intake, and similar results were obtained for energy, folates, calcium, iodine, zinc and magnesium. In particular, a very low vitamin D intake was reported-around 22% of the recommended intake- in both CD and non-CD subjects, without differences between males and females. Plasma levels of vitamin D were found between 10 and 30 ng/mL in 35% of CD patients. These results were not different when compared to the control group. These results strengthen the need for the measurement of circulating levels of vitamin D, in order both to select patients with vitamin D deficiency and to address them to a nutritional intake evaluation.

The age of the patients becomes crucial: a correct suggestion should consider how menopause age is drawing closer. In patients younger than 45 years, before the perimenopausal period, the persistence of vitamin D deficiency or its recurrence should be investigated towards, first of all, the identification of incorrect adherence to GFD, and secondly, the identification of other conditions responsible for this alteration. In Table 1, autoimmune conditions associated with CD and considered at high risk of osteoporosis are reported.

Under this light, one of the main actors of this coexisting alteration is represented by a thyroid disorder [29], which is very frequently associated to CD. In patients with Hashimoto’s thyroiditis and Grave’s disease, reduced levels of vitamin D were shown [55,62]. Vitamin D deficiency is considered to be responsible for the impaired T-cell suppression causing the release of proinflammatory cytokines and the consequent inflammatory-related alteration of thyroid structure, and therefore, function [63]. Many immune cells, such as macrophages, lymphocytes and dendritic cells, express vitamin D receptors and may convert 25-vitamin D into 1,25-vitamin D [64], which inhibits Th1 cell proliferation and Th1 cell cytokine production and decreases HLA class II antigen surface expression and B cell apoptosis [62,64,65,66]. Oral supplementation of 2000 IU/day vitamin D (more than twofold higher than RDA) is associated with a reduction in antithyroid autoantibodies suggesting a protective effect [67]. Moreover, it should be emphasized that in CD patients, intestinal malabsorption frequently causes iron deficiency, worsening thyroid function due to the impairment of heme-dependent TPO [68].

Another condition to rule out, if alterations of calcium balance are not dependent on gluten toxicity, is sarcoidosis [69]. It was recently reported by a systematic review and meta-analysis that the risk of sarcoidosis in CD patients is higher than subjects without CD with a pooled OR higher than 7 [70].

## 5. Conclusions

Alterations of bone and mineral metabolism are very frequent in CD patients and in comparison with the general population; the risk of fractures is significantly higher in this condition. Accordingly, besides a prompt diagnosis of CD, a prompt evaluation of bone mineral density and calcium metabolism is also mandatory in newly diagnosed patients to prevent the consequences of bone derangement and reduce the risk of fractures. A strict adherence to GFD proved to be effective in the reduction of risk fractures, even if in patients diagnosed after the achievement of the peak of bone mass, the reversal of bone loss is more difficult, in comparison with patients starting GFD in younger age.

The diet-induced improvement of calcium absorption, after the reversal of intestinal mucosa lesions, represents an important mechanism determining bone mineral density increase and making calcium supplementation less urgent. The need for vitamin D supplementation should be accurately evaluated because the beginning of a strict GFD may be rapidly and effectively adequate to revert calcium balance abnormalities detected at diagnosis, and consequently, to normalize serum levels of both 25-vitamin D and 1,25-vitamin D.

The relapse of alterations of calcium balance in patients on long-term GFD may represent a clinical sign of low adherence to GFD or the occurrence of another condition responsible for vitamin D deficiency. In this latter case, the mere correction of calcium balance will hesitate in a delay of an undiagnosed condition, together with patient exposition to its consequences and complications. Thyroid function should be promptly evaluated in case of vitamin D deficiency relapse.

The research agenda to elucidate vitamin D status in CD is still full of topics: first of all, studies dealing with vitamin D status in postmenopausal CD patients on long-term GFD are needed to ascertain whether this peculiar period of women physiology modifies the indication to vitamin D supplementation. Moreover, information is needed on the effect of vitamin D on immune response in CD patients, analyzing a possible effect on the RANK/RANKL/OPG system, known to be involved in the pathophysiology of bone loss. Finally, some information is needed on the pleiotropic effect of vitamin D, involving many endocrine and immune pathways, in relation of a possible proresorptive effect of the imbalance of the activity of other glands.

## Figures and Tables

**Figure 1 metabolites-13-00074-f001:**
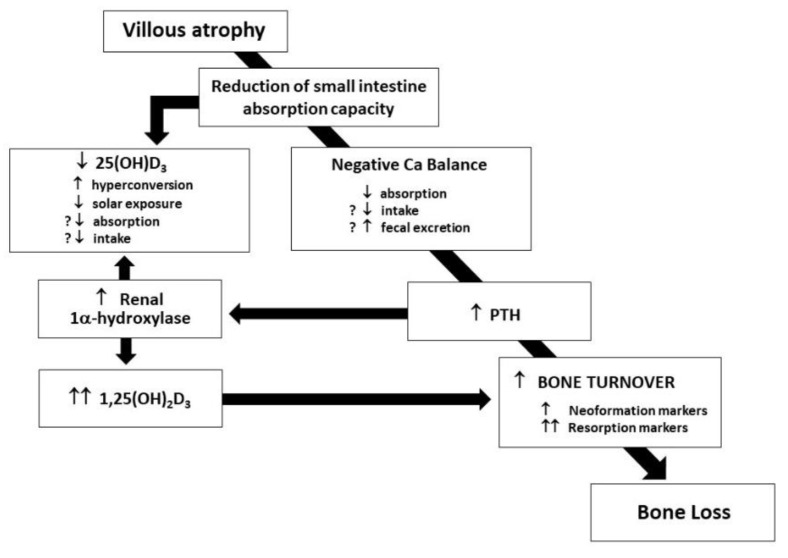
Alterations of calcium balance in untreated CD patients.

**Table 1 metabolites-13-00074-t001:** Autoimmune conditions associated with CD and considered at risk of osteoporosis.

Autoimmune Conditions	Vitamin D Status	Bone Loss
Hashimoto’s thyroiditis	deficiency	yes
Grave’s disease	deficiency	yes
Type 1 diabetes	deficiency	yes
Autoimmune liver diseases	deficiency	yes
Sjogren syndrome	deficiency	yes
Rheumatoid arthritis	deficiency	yes
Systemic lupus erithematosus	deficiency	yes
Sarcoidosis	deficiency	yes
Anti-phospholipid syndrome	deficiency	unknown
Williams–Beuren syndrome	unknown	yes
Addison’s disease	deficiency	yes
Autoimmune atrophic gastritis	deficiency	yes

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
