# Peer review of "The Effect of a Gluten-Free Diet on Vitamin D Metabolism in Celiac Disease: The State of the Art"

_metabolites, 2023, doi:10.3390/metabo13010074_

Round 1

Reviewer 1 Report

There are a number of paragraphs in the review where the information is correct, but it appears the text was written in another language and the English translation is awkward to read. Otherwise, the review is very good.

Examples:
Pg 1 Lines 26-28 - The sentence seems to have multiple tenses of verbs: it is not clear whether the verb "increases" refers to the immune response evoked by gluten or the determining of villous atrophy, etc.

Pg 1 Line 35 - should be resorption, not resportion

Pg 2 Lines 48-50 - Awkward wording, passive voice

Author Response

we thank the reviewer for its comments and we have now modified the text according to his suggestions.

Reviewer 2 Report

In my viewpoint, this review focussed in the evidences describing the impairment of bone and mineral metabolism and we have suggested a correct timing for their measurement and a correct strategy to select patients who need a vitamin D supplementation along their life.

I suggested several changes to clarify it:

1. Figure is cited in the manuscript, but should be corrected as Figure 1 in the text and figure caption

2. In line 288 should be numbered the Table as Table 1. Furthermore, table caption should be as Table 1. 

Author Response

we thank reviewer 2 for his comments and we have now modified the text according to his suggestions.

Reviewer 3 Report

The title should include a study type. 

In the introduction please highlight the knowledge gap which your review would cover. 

In second part of the review please include subheadings for easier reading. 

It appears valuable to add a table with a summary of available studies on the impact of GFD on vitamin D metabolism. 

Please include also some recent studies. 

Cardo A, Churruca I, Lasa A, Navarro V, Vázquez-Polo M, Perez-Junkera G, Larretxi I. Nutritional Imbalances in Adult Celiac Patients Following a Gluten-Free Diet. Nutrients. 2021 Aug 21;13(8):2877. doi: 10.3390/nu13082877. PMID: 34445038; PMCID: PMC8398893.

Verma A, Lata K, Khanna A, Singh R, Sachdeva A, Jindal P, Yadav S. Study of effect of gluten-free diet on vitamin D levels and bone mineral density in celiac disease patients. J Family Med Prim Care. 2022 Feb;11(2):603-607. doi: 10.4103/jfmpc.jfmpc_1190_21. Epub 2022 Feb 16. PMID: 35360767; PMCID: PMC8963631.

Line 174 instead "apart for" use "apart from"

Author Response

We thank the Reviewer #3 for his comments. A point-by-point reply is provided

The title should include a study type.

We agree with the reviewer. The title is misleading because suggest an original study dealing with the effect of GFD. On the contrary, the paperi s a review and this should be evident from the title. Accordingly, we have now modified the title.

In the introduction please highlight the knowledge gap which your review would cover. 

We agree with the Reviewer. In the “Introduction” section the “aim” of the review should be clear. Accordingly, we have now added a sentence in this section, to explain this point.

In second part of the review please include subheadings for easier reading. 

We agree with the Reviewer. We have now added two subheadings in the second part of the paper.

It appears valuable to add a table with a summary of available studies on the impact of GFD on vitamin D metabolism. 

We disagree with the reviewer on this point. A table summarizing all the available studies on the effect of GFD on vitaminD will cause a large overlap with information provided in the text, making duplication of data inevitable. We prefer to expalin these results in the text, separating the effect on 25 vitaminD and 1,25 vitaminD in untreated and treated CD patients. Such a Table will run the risk of being extremely confusing due to the need of mantain separate the two forms of vitaminD and the subgroups of CD patients.

Please include also some recent studies. 

Cardo A, Churruca I, Lasa A, Navarro V, Vázquez-Polo M, Perez-Junkera G, Larretxi I. Nutritional Imbalances in Adult Celiac Patients Following a Gluten-Free Diet. Nutrients. 2021 Aug 21;13(8):2877. doi: 10.3390/nu13082877. PMID: 34445038; PMCID: PMC8398893.

Verma A, Lata K, Khanna A, Singh R, Sachdeva A, Jindal P, Yadav S. Study of effect of gluten-free diet on vitamin D levels and bone mineral density in celiac disease patients. J Family Med Prim Care. 2022 Feb;11(2):603-607. doi: 10.4103/jfmpc.jfmpc_1190_21. Epub 2022 Feb 16. PMID: 35360767; PMCID: PMC8963631.

We thank the Reviewer for his suggestion. However, we have added the first paper (Cardo et al), but we have not added the second one: we excluded this paper during the writing of the review because it does not specify which form of vitamin D was measured (25 vitaminD or 1,25 vitaminD?).

Line 174 instead "apart for" use "apart from"

We apologize for the mistake. We have now correct this point.